# A Comparative Analysis on the Biochemical Composition and Nutrition Evaluation of Crayfish (*Procambarus clarkii*) Cultivated in Saline-Alkali and Fresh Water

**DOI:** 10.3390/foods14111997

**Published:** 2025-06-05

**Authors:** Yanqing Huang, Pengcheng Gao, Duanduan Yu, Zhen Sun, Xu Yang, Qifang Lai, Hai Chi

**Affiliations:** 1Key Laboratory of Inland Saline-Alkaline Aquaculture, Ministry of Agriculture and Rural Affairs, East China Sea Fisheries Research Institute, Chinese Academy of Fishery Sciences, Shanghai 200090, China; huangyq@ecsf.ac.cn (Y.H.); gaopc@ecsf.ac.cn (P.G.); sunzhen@ecsf.ac.cn (Z.S.); yangxu@ecsf.ac.cn (X.Y.); 2Key Laboratory of Protection and Utilization of Aquatic Germplasm Resource, Liaoning Ocean and Fisheries Science Research Institute, Dalian 116023, China; 15924494718@163.com

**Keywords:** *Procambarus clarkii*, quality analysis, nutrition profiles, saline-alkali aquaculture, amino acid composition, flavor compounds

## Abstract

To compare the quality and nutritional differences of crayfish under freshwater and saline-alkali water aquaculture conditions, this study systematically analyzed the biochemical composition, physical properties, and nutritional evaluation of crayfish (*Procambarus clarkii*) cultivated in saline-alkali water (PC-SW) and freshwater aquaculture (PC-FW), respectively. The results showed that crayfish from PC-SW had higher crude protein, crude fat, water content, and ash content. At the same time, PC-SW had a higher meat yield (16.18 ± 0.74%) than PC-FW (*p* < 0.05), with no significant changes in the head weight ratio or hepatopancreas weight ratio, indicating superior crayfish quality. The trace element content of PC-SW differed significantly from that of PC-FW (*p* < 0.05), with the exception of Cu. To some extent, the amino acid and fatty acid compositions were similar. The no essential amino acids content of crayfish cultivated under freshwater and saline-alkali conditions was higher than the essential amino acids content. The total branched-chain amino acids (BCAAs) content was higher than the total aromatic amino acids (AACs) content in both groups; however, the BCAA to AAC ratio was similar, at approximately 2.14. The essential amino acid index results were 69.01 and 68.02, respectively. Finally, betaine and nucleotide concentrations increased and geosmin content was significantly reduced in PC-SW (3.13 ± 0.09 μg/kg) compared to PC-FW (4.32 ± 0.09 μg/kg) (*p* < 0.05), implying that PC-SW crayfish had a better flavor. Our findings revealed that cultivating crayfish under saline-alkali conditions can significantly improve the nutritional quality and flavor of muscle.

## 1. Introduction

*Procambarus clarkii*, commonly known as freshwater crayfish, belongs to the *Crustacea* order, *Decapoda* family, and *Procambarus* genus. Crayfish is native to northern Mexico and the southern and southeastern United States. Crayfish was first introduced to China in the 1930s and is highly popular because of its tender and delicious flesh [1,2]. China now is the largest producer of crayfish worldwide. According to the “China Crayfish Industry Development Report (2024)”, China’s crayfish aquaculture production reached 3.161 million tons in 2023, accounting for the great majority of worldwide crayfish aquaculture production [3,4]. In 2003, the Chinese market for crayfish was valued at roughly 68.2 billion US dollars [3,4].

Crayfish are highly adaptable and resistant to stress, allowing them to live in various environments [5]. They have substantial tolerance to hypoxic conditions and can survive in environments where other fish struggle to thrive. As a result, crayfish have the potential to be introduced into saline-alkali water for aquaculture purposes.

China has 9.9 × 10^11^ m^2^ of saline-alkali land, which accounts for 1.03% of the national land area [6]. Low-lying saline-alkali water covers up to 4.6 × 10^11^ m^2^, or 55% of the total lake area in China [6,7]. Most water sources are abandoned because of their excessive salinity and alkalinity, which prevent them from being ingested by humans and animals, and are used for agricultural irrigation. Therefore, accelerating the development and utilization of saline-alkali water resources, as well as establishing saline-alkali water aquaculture, is critical for enhancing the ecological environment and fishery economy [6,8,9]. In recent years, several studies have used saline-alkali water to establish a saline-alkali fishery industry system. In addition, some fish species, such as *Penaeus vannamei*, Tilapia, and mud crab, have been shown to exhibit high survival and physiological adaptation in saline-alkali water [7,10,11]. Weight gain, growth rate, survival rate and feed conversion ratio of *P*. *vannamei* significantly changed in a salt-alkali culture system compared to a freshwater one [7]. After aquaculture, they may reduce water salinity, enhance the ecological environment, and boost fishery productivity and farmer revenue [11].

Saline-alkali water is deficient in ions and has restricted availability. Fish in saline-alkali environments require extra energy for ion transport to maintain the acid–base and osmotic balances in their bodies [7]. When osmotic pressure fluctuates in fish, it affects a variety of physiological metabolisms, osmotic managements, and gut microbiota, all of which influence the synthesis and breakdown of proteins, lipids, and other substances in the muscles [7,12,13,14]. Current studies indicate that aquatic animals cultivated under saline or alkaline water conditions have improved muscle nutrition and quality. Qin et al. [10] also reached the same conclusion, finding that saline-alkali water can increase the content of free amino acids and nucleotides in the muscles of *Litopenaeus vannamei*, improving the overall flavor of the muscles. Jia et al. [15] found that 0.5% and 0.75% salinity can improve muscle texture, increase the content of amino acids and fatty acids, and minimize soil taste and flavor components in grass carp (*Ctenopharyngodon idella*) and black carp (*Mylopharyngodon picyma*).

With the continuous improvement in living standards, the demand for aquatic products has shifted from quantity to quality. Consumers are increasingly interested in the flavor, texture, nutrition, safety, and other aspects of aquatic products [16]. Currently, there are several studies on the analysis and quality evaluation of muscle nutrition components in crayfish, but there are few studies on the nutrition quality of crayfish cultivated in saline-alkali water. This study aimed to investigate the nutritional content and qualitative properties of crayfish cultivated from saline-alkali and fresh water systematically. Based on the analysis of muscle biochemical composition, trace element composition, muscle dehydration rate, stream loss, nucleotides, betaine, geosmin, amino acid composition, and fatty acid composition in crayfish cultivated in fresh and saline-alkali water, our findings could provide a scientific basis for future crayfish and feed development in saline-alkali water aquaculture. On the other hand, these findings also can improve the nutritional value of crayfish and serve as a foundation for residents’ rational consumption and scientific selection of aquatic products.

## 2. Materials and Methods

### 2.1. Ethical Statement

Animal experiments were approved by the Ethics Committee of the East China Sea Fisheries Research Institute, Chinese Academy of Fishery Sciences (Approval code: 2024-12-ZX-013).

### 2.2. Sample Collection

Juvenile crayfish (2.0 ± 0.1 cm in length, 5.0 ± 0.3 g in weight) from the same batch and farm were obtained in May 2024 from Qingjiang, Hubei province, and approximately 1000 juvenile crayfish were distributed under freshwater (PC-FW) and saline-alkali (PC-SW) aquaculture conditions. The freshwater and saline-alkali aquaculture regions are located in Qianjiang (Hubei province) and Shahu (Ningxia Hui Autonomous Region), respectively. Each aquaculture farm underwent a water change from the local well water every 15 days, with each change typically accounting for a quarter of the total farm water volume, to avoid excessive fluctuations in water quality that may affect crayfish growth. After each water change, the salinity, alkalinity, and pH were measured accordingly. The pH value was detected by using a portable pH meter (TOH01139, Shenzhen Yier Scientific Instrument Co., Shenzhen, China). The salinity and alkalinity were detected by using partial portable conductance (SevenDirect SD30, Mettler Tolede, Zurich, Switzerland). The salinity, alkalinity, and pH under saline-alkali conditions are listed in Appendix A. After three months of domestication under these two breeding conditions, 50 crayfish that displayed elevated vitality and an undamaged appearance were selected and transported to the laboratory in low-temperature ice packs for future use. The crayfish were thoroughly washed under running water to remove the surface debris and silt. After weighing and draining, the head, internal organs, and shell of the crayfish were removed, and the crayfish flesh was used for further examination.

### 2.3. Determination of Meat Yield (MY) of Crayfish

The meat yield (MY) of the crayfish was determined using the method described by Lv et al. [17]. Briefly, the head and claws of the crayfish were first removed, followed by the abdomen, which was cut from the line along both sides of the crayfish. Subsequently, the muscles were removed using a spoon. The MY rate was calculated following Equation (1).(1)$$MY\;(\%)=\frac{m_{0}}{m_{1}}\times 100\%$$ where m_0_ means muscle weight (g) taken from the abdomen of crayfish, and m_1_ means total (g) weight of crayfish.

### 2.4. Determination of Nutritional Components in Crayfish

Nutritional components are reflected by water content, crude protein content, crude fat content, and ash in this study based on the method described in a previous study [18]. Water content in the crayfish was measured by using 5.00 ± 0.05 g of crayfish flesh in a moisture analyzer (Computrac MAX 4000XL, AMTEK-Brookffeld, Middleborough, MA, USA). The crude protein content was tested by using the Kjeldahl method. Approximately 1.20 ± 0.03 g of crayfish flesh was homogenized (T18, IKA, Staufenim, Germany). The samples were used for crude protein content detection in an automated Kjeldahl apparatus (HGK-50, Chaosheng, Shanghai, China) and ash content detection in an ash analyzer (Rapid continuous ash analyzer, HF-2, Henan, China). All the tested indexes of nutritional components of crayfish were followed by Khan et al. [18].

### 2.5. Determination of Trace Elements in Crayfish

Trace elements, including copper (Cu), ferric (Fe), zinc (Zn), cesium (Se), potassium (K), sodium (Na), calcium (Ca), magnesium (Mg), and manganese (Mn) in the crayfish, were detected based on the method described by Chi et al. [19]. Briefly, approximately 0.50 ± 0.01 g of homogenized crayfish flesh was mixed with 5 mL of nitric acid (2%) and kept at room temperature for 1 h. The samples were digested in a microwave digesting instrument (TOPEX, Shanghai, China) using the following programs: 100 °C for 3 min, 140 °C for 3 min, 160 °C for 3 min, 180 °C for 3 min, and 190 °C for 15 min. After digestion, the samples were cooled down to 50 °C and washed 3 to 4 times with ultrapure water. The samples were fixed in a flask at 25 mL. The trace elements in crayfish after the samples passed through a 0.45 μm filter membrane were determined using an atomic absorption spectrophotometer (ICP-OES optima 8000, Waltham, MA, USA). The parameters for ICP-OES were 5 L/min auxiliary gas flow rate, 12 L/min cooling flow rate, and 5 mm sample depth. The method validation results are shown in Appendix A.

### 2.6. Determination of Muscle Dehydration Rate (MDR) and Steaming Loss Rate (SLR) in Crayfish

The muscle dehydration rate (MDR) and steaming loss rate (SLR) were determined based on the method described by Lv et al. with a slight modification [17]. Briefly, approximately 5.00 ± 0.02 g of crayfish was cut into small pieces and wrapped with qualitative filter paper. The sample was placed in a centrifuge tube and centrifuged at 3500× *g* for 10 min. The samples were then removed and accurately weighed. The MDR was calculated after the sample weight, according to Equation (2).(2)$$MDR\;(\%)=\frac{1-\left(m_{0}-m_{1}\right)}{m_{0}}\times 100\%$$ where M_0_ is the muscle weight (g) of crayfish before centrifugation and M_1_ is the muscle weight (g) of the centrifuged crayfish.

The cut crayfish muscles were placed in a flask and soaked in a water bath at 85 °C for 15 min. The samples were weighted after water was completely absorbed. The SLR was calculated according to the Equation (3).(3)$$SLR\;(\%)=\frac{m_{0}-m_{1}}{m_{0}}\times 100\%$$ where M_0_ is the muscle weight (g) of crayfish prior to the water bath and M_1_ is the muscle weight (g) of crayfish after the water bath.

### 2.7. Analysis of Amino Acid Composition in Crayfish

The amino acid composition of the crayfish was determined in accordance with Zhou et al. [20]. About 30.00 ± 0.01 mg of sample was mixed with 15 mL of 6 mol/L HCl and transferred to a nitrogen-filled hydrolysis tube. The mixture was hydrolyzed for 24 h at 110 °C in a drying oven before being fixed to 50 mL with ddH_2_O. The hydrolysate was concentrated in a vacuum and then dissolved with 1 mL of 4 mol/L HCl. The mixture was filtered through a 0.45 μm membrane and tested for amino acids using an amino acid analyzer (LA8080, Hitachi Co., Tokyo, Japan).

### 2.8. Analysis of Fatty Acid Composition in Crayfish

The fatty acid composition of the crayfish was measured using the method described by Ran et al. [21]. Approximately 2.00 ± 0.01 g of crayfish muscle was mixed with 5 mL of 2% methanol sulfate. Following transesterification at 70 °C for 1 h, approximately 2 mL of the supernatant was thoroughly mixed with 0.75 mL of ddH_2_O and 2 mL n-hexan. A total of 1 mL of the filtered sample after stratification through a 0.45 μm membrane was used for fatty acid composition determination via gas chromatography (Trace1310 ISQ, Thermo Scientific, Waltham, MA, USA). The test parameters were as follows: HP-88 column 100 m × 0.25 mm × 0.20 μm, injection port temperature 260 °C, carrier gas flow rate 1.0 mL/min, split ratio 20:1, ion source temperature 250 °C, and transmission line temperature 280 °C, solvent delay time 5 min, and EI 70 eV as ion source.

### 2.9. Determination of Betaine in Crayfish

Betaine in the crayfish was detected using the method described by Servillo et al. [22] with slight modifications. Briefly, samples (0.50 ± 0.02) g were combined with 6% perchloric acid and gently shaken for 5 min. After 20 min of ultrasonication, the mixture was centrifuged at 7000× *g* for 10 min. After centrifugation, the supernatant of the mixture was transferred to a 10 mL tube and the pH was adjusted to 6.5, using 50 mmol/L of NaOH. Finally, the mixture was fixed to 3 mL with ultrapure water and filtered through a 0.45 μm membrane. Betaine was detected using approximately 1 mL of the combined sample by high-performance liquid chromatography (Kontron PC with Inear Uvis 200 detector, Kontron HPLC pump 422, Eching, Germany). The detector column was TSK GEL ODS-100V (4.6 × 250 mm, 0.5 μm). The mobile phase was acetonitrile: water (60:40) and the detection wavelength was 195 nm.

### 2.10. Determination of Nucleotides in Crayfish

Nucleotides in the crayfish were evaluated as previously described by Zhao et al. [23] with slight modifications. The sample (2.00 ± 0.01 g) was ultrasonically dissolved in 5 mL of 20% perchloric acid for 10 min. The mixture was then centrifuged at 8000 rpm for 15 min at 4 °C. This procedure was repeated using 5% perchloric acid. The supernatants were blended, neutralized (pH 6.5) with 1 mol/L KOH, fixed to 50 mL with ddH_2_O, percolated through a 0.45 μm filter membrane, and eventually analyzed by high-performance liquid chromatography (Agilent 1260, Agilent Technologies Inc., Santa Clara, CA, USA).

### 2.11. Evaluation of Nutritional Quality of Crayfish

Nutritional quality was expressed by the amino acid score (AAS), chemical score (CS), and essential amino acid index (EAAI) in this study based on the recommended scoring criteria for amino acids per gram of nitrogen by the FAO/WHO. The AAS, CS, and EAAI were calculated according to Equations (4)–(6).(4)$$AAS=\frac{A_{0}}{A_{1}}$$ where A_0_ is the content (mg/g) of a certain amino acid in crayfish and A_1_ is the content (mg/g) of the same amino acid in the FAO/WHO standard model.(5)$$CS=\frac{A_{0}}{A_{2}}$$ where A_0_ is the content (mg/g) of a certain amino acid in crayfish and A_2_ is the content (mg/g) of homologous amino acids in whole egg protein.(6)EAAI=A0T0×100×A1T1×100×......×AnTn×100n where n is the number of amino acids, A_0_, A_1_, ..., and A_n_ are the contents (mg/g) of various amino acids in crayfish, and T_0_, T_1_, ..., and T_n_ are the contents (mg/g) of various amino acids in whole egg protein, respectively.

### 2.12. Geosmin Detection

The geosimin content from both PC-SW and PC-FW groups was detected as described by Brisow et al. [24].

### 2.13. Data Analysis

Data from at least three independent repeated parallelisms are presented as the mean ± standard deviation. Data were analyzed using one-way ANOVA in SPSS 20.0 (IBM, Chicago, IL, USA) to determine significance. Duncan’s method was used to evaluate the groups, and *p* < 0.05 indicated significant differences.

## 3. Results and Discussion

### 3.1. Nutritional Profiles of Crayfish

Figure 1 depicts the nutritional profiles of crayfish in freshwater and saline-alkali water. The nutritional characteristics of PC-SW were determined to be 18.01 ± 0.05% crude protein, 0.71 ± 0.06% crude fat, 79.38 ± 1.01% water content, and 1.60 ± 0.17% ash. All the tested data were higher than those from PC-FW. However, there were significant differences (*p* < 0.05) in crude protein and ash levels between the two groups. Our study found that crude protein and crude fat content in crayfish increased under saline-alkali conditions. This conclusion might be attributed to the promotion of endogenous synthesis of certain fatty acids under saline-alkali conditions, resulting in an increase in fat content. Similar findings have been reported in studies on *Lateolabrax japonicus* and *Micropterus salmoides*, as well as in *Scophthalmus maximus*, where the crude fat content decreased as salinity increased [25,26,27]. Several studies have investigated the nutritional value of crayfish in freshwater environments [28,29]. Lazarevic et al. found that invasive crayfish in Serbia showed comparable protein levels (18.12%), ash (1.37%), and fat content (0.25%) to our study [28]. Sun et al. found that crayfish had higher nutritional profiles in terms of protein, ash, and water content, which might be contributed to the breeding regions and conditions [30]. Related studies have also revealed that nutritional profiles may be associated with sampling months. Crayfish mate and lay eggs mostly between April and May, when their protein and fat levels decline. The results implicitly indicate the gap in protein and fat contents between prior studies.

### 3.2. Meat Yield, MDR, and SLR in Crayfish

Meat yield is an indicator of the quality of aquatic products, parental germplasm, and economic features. Additionally, meat yield is an important factor in determining crayfish consumption because its main edible part is the tail itself. The results of meat yield, MDR, and SLR in crayfish are shown in Table 1. PC-SW exhibited a higher meat yield (13.41 ± 1.51%) than PC-FW (16.18 ± 0.74%). No significant differences were found in MDR between the two groups, while PC-SW presented a higher MDR content (81.89 ± 0.60%). In contrast, the SLR contents of PC-SW and PC-FW were 12.10 ± 1.50% and 23.93 ± 2.00%, respectively. Based on this evaluation criterion, crayfish quality in PC-SW was higher than that in PC-FW. However, the fact that PC-SW exhibited a higher meat yield than PC-FW could be attributed to salt-alkali stress or salt-alkali applicability.

Individual characteristics, growth environment, and nutrient intake can potentially influence meat yield. Sun et al. found that male redclaw crayfish showed a higher meat yield (19.21 ± 0.61%) than females (15.81 ± 0.36%). The meat yield in our study was lower than that of the redclaw crayfish, which may be attributed to the well-developed limbs and thick shells of crayfish. Our findings are in agreement with those reported by Chen et al. [31]. In addition, PC-SW had a lower SLR content than PC-FW. The higher concentration of Na in saline-alkali environments may compete with K to impede its uptake and retention in the muscle. Because K is a crucial ion that helps maintain the structure of myofibrillar forces between proteins, muscle fibers become more prone to fracture and release water when heated [32]. This could be the reason for the substantial disparities in the SLR. Future research should concentrate on the structural alterations in crayfish under various breeding conditions.

### 3.3. Trace Elements in Crayfish

The results of contents of trace elements in PC-SW and PC-FW showed that the content of Cu in PC-FW (34.68 ± 0.30 mg/kg) was higher than that in PC-SW (32.48 ± 0.22 mg/kg) (shown in Figure 2). However, under saline-alkali conditions, the content of the other eight trace elements in PC-SW was higher than that of trace elements in PC-FW. Significant differences were found on the concentrations of Fe, Zn, Se, K, Na, Ca, Mg, and Mn in PC-SW, with respective concentrations of 102.49 ± 1.69 mg/kg, 95.45 ± 2.00 mg/kg, 2.49 ± 0.04 mg/kg, 18.29 ± 0.06 g/kg, 8.66 ± 0.01 g/kg, 6.09 ± 0.01 g/kg, 2.28 ± 0.01 mg/kg, and 18.38 ± 0.20 mg/kg. Accordingly, the concentrations of Fe, Zn, Se, Na, Ca, Mg, and Mn in PC-FW were 24.46 ± 0.07 mg/kg, 35.16 ± 0.30 mg/kg, 1.20 ± 0.00 mg/kg, 15.83 ± 0.52 g/kg, 4.19 ± 0.14 g/kg, 1.12 ± 0.03 g/kg, 1.52 ± 0.04 g/kg, and 11.56 ± 0.02 g/kg, respectively. Lazarevic et al. [28] analyzed the macrominerals in the crayfish (*Faxonius limosus*), and their results showed that K was the most prevalent in the meat, followed by Na, Ca, and Mg. These results indicate that varying trace element concentrations likely result in different levels of environmental adaptation. Future work could consider tracing trace elements and their interactions with environmental factors.

Related studies have shown that trace elements are important components of aquatic products, affecting not only metabolism, growth and development, and disease prevention, but also the nutritional value and flavor of aquatic meat, such as Cu, Mn, and Fe, which can cause unsaturated lipid oxidation [33,34,35]. Additional studies have shown that Mg, Ca, Fe, K, and Na ions have a salty taste, Zn ions exhibit a sweet taste, and Cu and Mn ions exhibit a bitter taste. Hence, the flavor of aquaculture meat is greatly influenced by these trace elements [34].

### 3.4. Fatty Acids Profiles in Crayfish

In our study, 23 fatty acids in the muscles of PC-FW and PC-SW, including saturated fatty acids (SFAs), monounsaturated fatty acids (MUFAs), and polyunsaturated fatty acids (PUFAs), were examined (Table 2). The results showed that the total SFA content in PC-SW (32.72%) was significantly greater than that (29.68%) in PC-FW (*p* < 0.05), while the total PUFA content in PC-FW showed higher values at 39.12%, which was reflected in the enrichment of n-3 fatty acids, such as C18:3n3 and C20:4n6 (ARA). In contrast, PC-SW had 14.06% C20:5n3 (EPA), whereas PC-SW had 12.16% EPA. Meanwhile, both groups showed an n-3/n-6 PUFA ratio greater than 1.4, indicating their high nutritional value.

Dynamic changes in the fatty acid profiles of PC-SW and PC-FW also demonstrate quality and nutritional differences. Our findings found that C16:0 exhibited the largest content among SFA, whereas C14:0 exhibited the lowest level, with concentrations of 14.45 ± 0.42 g/100 g and 16.69 ± 0.63 g/100 g, respectively. Saline-alkali conditions may promote the biosynthesis of more SFA, such as C16:0, to maintain cell membrane fluidity and to adapt to high salt pressure. Meanwhile, desaturase acts on C16:0 to produce C16:1n7, which the PC-FW group may accumulate because of decreasing C16:0 levels or increased enzyme activity [36]. This may also explain why C16:1n7 in the PC-FW group was higher than that in the PC-SW group in the MUFA. The ∑n-3 PUFA content of the PC-FW group was slightly higher than that of the PC-SW group, with C18:3n3 (LNA) being significantly higher in the PC-FW group than in the PC-SW group. Under saline-alkali conditions, crayfish may more efficiently convert LNA into EPA, resulting in a higher EPA content in the PC-SW group, while the PC-FW group accumulated more precursors [37]. Finally, the EPA and total highly unsaturated fatty acid (HUFA) contents of the PC-SW group were higher, consuming crayfish from saline-alkali conditions; therefore, it may be more beneficial for the cardiovascular health of consumers. Future studies should focus on this area of work.

### 3.5. Amino Acid Composition and Content in Crayfish

As shown in Table 3, 17 amino acids were identified in the muscles of both saline-alkali water and freshwater crayfish, including eight essential amino acids (EAAs), two semi-essential amino acids, and seven non-essential amino acids (NEAAs). Interestingly, there were no significant differences in amino acid composition between the two groups. The total amino acids (TAAs), total EAAs, and total delicious amino acids (TDAAs) for PC-FW were 78.88 ± 0.47 g/100 g, 29.34 ± 0.18 g/100 g, and 76.70 ± 0.46 g/100 g, 28.65 ± 0.18 g/100 g for PC-SW, respectively. Therefore, the ratios of TDAAs to TAAs and EAAs to TAAs for the two groups were approximately 39% and 37%, respectively.

The type and composition of amino acids have a significant impact on the growth and development of aquatic species as well as the evaluation of their protein quality [38]. The human body requires EAAs from external food; therefore, the types, quantities, and composition ratios of EAAs are key indicators that determine the nutritional value of protein in food. There was no significant difference in EAAs, NEAAs, umami amino acids, or TAAs in the muscles of the two crayfish breeding modes. It can be seen that the types and quantities of amino acids in crayfish muscles are mostly regulated by factors such as breed and heredity, with little effect from breeding mode. This is consistent with the research findings of Zhang et al. [29] and Sun et al. [30].

### 3.6. Analysis of AAS, CS, and EAAI in Crayfish

AAS, CS, and EAAI in both PC-FW and PC-SW were evaluated against the FAO/WHO standard and whole egg protein (Table 4). Both PC-FW and PC-SW exhibited lower total EAA contents (2130 mg/g and 2126 mg/g, respectively) than the whole egg standard (3059 mg/g), whereas they corresponded more closely with the FAO/WHO standard. Notably, the Met+Cys levels in both concentrates (139 mg/g for PC-FW and 136 mg/g for PC-SW) were lower than the FAO/WHO standard (220 mg/g), indicating a potential nutritional constraint. In contrast, the Trp content significantly exceeded the standard in PC-SW (121 mg/g), with a remarkably high AAS of 2.02, suggesting greater bioavailability. Lys and Phe+Tyr likewise demonstrated favorable profiles, with AAS values above 1.0 in both concentrates, suggesting adequate nutritional sufficiency.

CS and EAAI further revealed the differences between the two groups. PC-FW and PC-SW showed similar total AAS and CS values, with PC-FW showing total AAS and CS of 5.61 and 7.86, respectively, and PC-SW showing 5.84 and 8.24, respectively. PC-SW exhibited significant differences in EAAI (*p* < 0.05), indicating an improved overall amino acid balance. The concentrations of Leu and Ile in both groups approached the FAO/WHO requirements, with the AAS values exceeding 0.9. However, Val and Thr displayed poor CS values of 0.55 and 0.72 for PC-FW, 0.53 and 0.70 for PC-SW, respectively, suggesting potential areas for formulation enhancement. These findings emphasize the variability in protein quality between concentrates, as well as PC-SW’s slight nutritional advantage over PC-SW, particularly in Trp and Lys content, which may enhance its suitability for dietary applications requiring tailored amino acid supplementation.

The amino acid content of aquatic products has a significant impact on their nutritional value and serves as an important indicator of their quality [39]. EAAI values can be used to quantify the degree of similarity between EAAs and standard protein content. EAAI above 0.95 indicates a high-quality protein source; EAAI between 0.86 and 0.95 indicates a good protein source; EAAI between 0.75 and 0.86 indicates an available protein source; and EAAI below 0.75 indicates an inappropriate protein source [40]. Our EAAI results for both the PC-SW and PC-FW groups were less than 0.75, indicating that both groups received protein from inappropriate sources. Chen et al. analyzed the muscle texture characteristics and nutritional quality of breeding freshwater crayfish under different feed conditions, and their findings on EAAI values were all above 0.95 [31]. These differences may be attributed to the breeding conditions and feed nutrition. Zhang et al. reported that the TAAs and EAAs in the muscle of crayfish (*Procambarus clarkii*) did not increase with dietary protein level, which may be related to amino acid equilibrium [41]. According to the CS score in our study, Met+Cys was the first limiting amino acid and Val was the second limiting amino acid in the muscle of crayfish, which is consistent with the results of crayfish in Dongting Lake, Changshu, *Penaeus vannamei*, and *Macrobrachium nipponense* described by Putra et al. [42] and Ge [43]. The findings revealed that shrimp species such as crayfish, *Penaeus vannamei*, and *Macrobrachium nipponense* from different origins, natural or aquaculture modes, generally lack oMet+Cys.

### 3.7. Betaine, Geosmin, and Nucleotide Content in Crayfish

Betaine and nucleotides, including CMP, UMP, and IMP, were higher in PC-SW than in PC-FW (shown in Table 5), with the exception of GMP, which was not detected in PC-FW. The results indicated that crayfish in saline-alkali cultivation conditions likely increased the concentration of betaine and nucleotides, particularly CMP and IMP, in PC-SW. CMP and IMP with respective concentrations of 172.20 ± 2.87 μg/g and 2835 ± 63.27 μg/g significantly differed from that of PC-FW with concentrations of 71.70 ± 0.41 μg/g and 1283.69 ± 43.79 μg/g, whereas the geosmin content differently decreased to 3.13 ± 0.09 ng/g in the PC-SW group, showing a significant difference from the PC-FW group. This difference may also imply that crayfish tastes better in saline-alkali environments than in freshwater environments. IMP is an umami enhancer that works synergistically with Glu to significantly enhance flavor intensity, creating a pleasant sensation. IMP may also provide ribose in the body to participate in the Maillard reaction and is an important precursor for producing meat flavor, indicating that IMP is the primary indicator of aquatic flavor. In our study, the IMP content in PC-SW was the highest among all nucleotides, which was 2.2 times that in PC-SW, indicating that IMP can be used as an essential quality indicator for comparing the quality of crayfish under fresh and saline-alkali water conditions.

Geosmin is mainly synthesized by actinomycetes such as *Streptomyces* sp. [44]. After cultivating crayfish in saline-alkali conditions, the pH of the water changes and salinity is regulated, which differs from the living environments of the original geosmin-producing strains. In addition, crayfish feces contain high levels of nitrogen and phosphorus, which promotes the large-scale reproduction of heterotrophic microorganisms that compete with actinomycetes for carbon sources and living conditions. Relevant research results have shown that the abundance of actinomycetes in water can be reduced by 30–50% under saline-alkali aquaculture conditions [45]. Therefore, the “fishing to reduce alkalinity” model could provide improved soil structure, safety water, and a reduction in the flavor of freshwater fish farming.

## 4. Conclusions

In this study, we conducted a comprehensive analysis of the biochemical and nutritional composition of crayfish (*Procambarus clarkii*) cultivated in fresh and saline-alkali water. It was found that there were certain differences in the basic nutritional composition, amino acids and fatty acids composition between freshwater and saline-alkali water aquaculture of crayfish, but most of the indicators were not significantly different. It is worth noting that the crayfish muscle cultivated in saline-alkali water was particularly rich in ions, making it a high-quality aquatic protein rich in K, Na, Ca, Mg, Zn, and Se. The flavor substance IMP in its muscle is extremely high, and the content of betaine is abundant. Furthermore, the content of geosmin is lower than that in freshwater cultured crayfish, implying that the flavor of crayfish cultured in saline-alkali conditions is more prominent than that of freshwater cultured crayfish. The findings obtained in this study showed that the cultivating crayfish in saline-alkali water can effectively improve its muscle flavor, not only increasing the scope of future cultivation of crayfish, but also expanding the suitable varieties for saline-alkali water cultivation. Due to the fact that there are three different types of saline-alkali land in China, the corresponding salinity, alkalinity, and pH are also different. Future studies should include laboratory validation based on the different saline-alkali concentration used by crayfish, as well as further exploration of its nutritional quality and other indicators.

## Figures and Tables

**Figure 1 foods-14-01997-f001:**
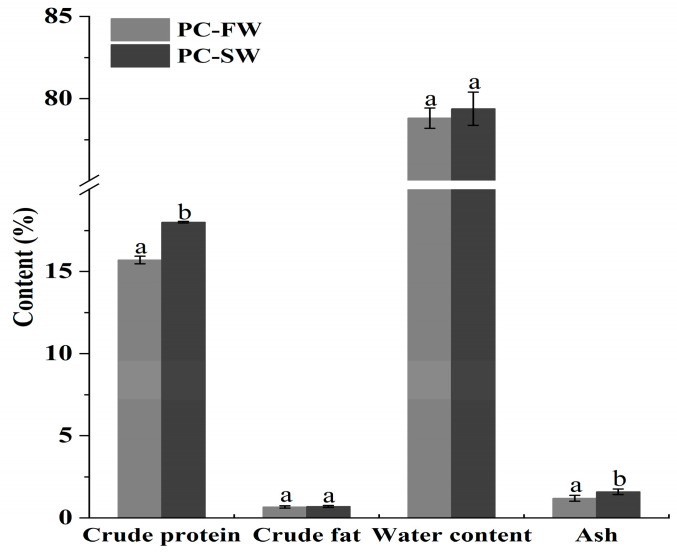
Nutritional contents of crayfish (fresh weight, %) in freshwater and saline-alkali water. Different superscripts on the same column of data indicate significant differences (*p* < 0.05).

**Figure 2 foods-14-01997-f002:**
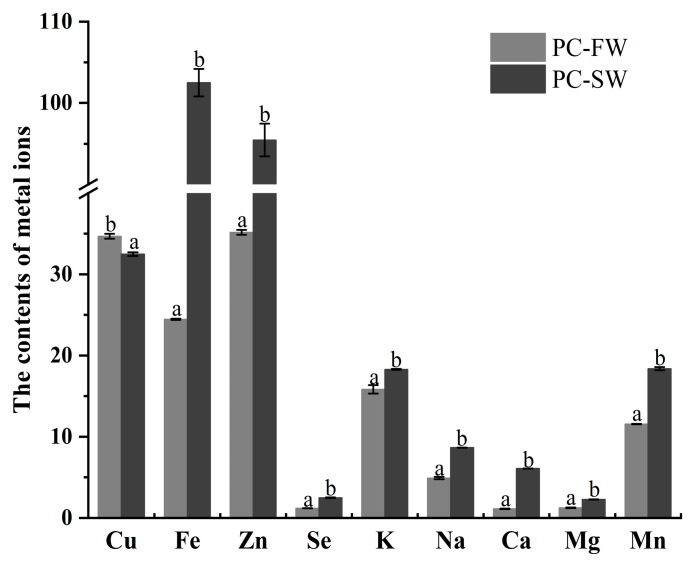
The contents of different trace elements of the crayfish muscles (dried weight) in freshwater and saline-alkali water. The concentration unit of K, Na, Ca, and Mg is g/kg, and the concentration unit for rest of the trace elements is mg/kg in this study. Different superscripts on the same column of data indicate significant differences (*p* < 0.05).

**Table 1 foods-14-01997-t001:** The results of meat yield, MDR, and SLR in crayfish in fresh and saline-alkali water (fresh weight, 100%).

	Meat Yield	Head Weight Ratio	Hepatopancreas Weight Ratio	MDR	SLR
PC-FW	13.41 ± 1.51 ^a^	74.52 ± 0.10 ^a^	5.97 ± 0.32 ^a^	79.93 ± 2.17 ^a^	23.93 ± 2.00 ^a^
PC-SW	16.18 ± 0.74 ^b^	75.21 ± 2.19 ^a^	3.69 ± 0.37 ^b^	81.89 ± 0.60 ^a^	12.10 ± 1.50 ^b^

Note: Different superscripts on the same column of data indicate significant differences (*p* < 0.05).

**Table 2 foods-14-01997-t002:** Fatty acid profiles in crayfish in fresh and saline-alkali water (g/100 g).

	PC-FW	PC-SW
C14:0	0.38 ± 0.06 ^a^	0.41 ± 0.06 ^a^
C15:0	1.47 ± 0.06 ^a^	1.73 ± 0.05 ^b^
C16:0	14.45 ± 0.42 ^a^	16.69 ± 0.63 ^b^
C17:0	0.40 ± 0.04 ^a^	0.46 ± 0.05 ^a^
C18:0	8.32 ± 0.15 ^a^	8.74 ± 0.08 ^a^
C20:0	0.67 ± 0.05 ^a^	0.70 ± 0.06 ^a^
C22:0	0.67 ± 0.04 ^a^	0.67 ± 0.10 ^a^
C23:0	3.31 ± 0.17 ^a^	3.32 ± 0.30 ^a^
ΣSFA	29.68 ± 0.34 ^a^	32.72 ± 0.35 ^b^
C14:1n5	0.03 ± 0.01 ^a^	0.03 ± 0.03 ^a^
C16:1n7	4.95 ± 0.19 ^b^	3.87 ± 0.45 ^a^
C17:1n7	0.02 ± 0.01 ^a^	0.02 ± 0.01 ^a^
C18:1n9	25.42 ± 0.23 ^a^	26.18 ± 1.02 ^b^
C20:1	0.55 ± 0.10 ^a^	0.56 ± 0.05 ^a^
C22:1	0.23 ± 0.09 ^b^	0.13 ± 0.02 ^a^
ΣMUFA	31.20 ± 0.25 ^a^	30.79 ± 1.43 ^a^
C18:2n6 (LA)	14.32 ± 0.49 ^b^	13.42 ± 1.29 ^a^
C18:3n3 (LNA)	4.65 ± 0.19 ^b^	2.46 ± 0.34 ^a^
C18:3n6	0.18 ± 0.01 ^a^	0.19 ± 0.04 ^a^
C20:2n9	0.95 ± 0.09 ^b^	0.73 ± 0.20 ^a^
C20:4n6 (ARA)	0.92 ± 0.04 ^b^	0.65 ± 0.14 ^a^
C20:5n3 (EPA)	12.16 ± 0.09 ^a^	14.06 ± 1.10 ^a^
C22:5n3 (DPA)	1.14 ± 0.32 ^a^	1.35 ± 0.18 ^b^
C22:6n3 (DHA)	3.78 ± 0.17 ^a^	3.63 ± 0.39 ^a^
ΣPUFA	39.12 ± 0.22 ^a^	36.49 ± 1.71 ^a^
Σn-3PUFA	22.74 ± 0.28 ^a^	21.50 ± 0.93 ^a^
Σn-6PUFA	15.24 ± 0.45 ^a^	14.06 ± 1.18 ^a^
n-3/n-6PUFA	1.49 ± 0.06 ^a^	1.53 ± 0.11 ^a^
ΣHUFA	70.32 ± 0.34 ^a^	67.28 ± 0.35 ^a^
DHA/EPA	0.29 ± 0.01 ^a^	0.26 ± 0.05 ^a^

Note: Different superscripts on the same column of data indicate significant differences (*p* < 0.05).

**Table 3 foods-14-01997-t003:** Composition and content of amino acids in muscle of crayfish (dried weight, g/100 g).

Amino Acids	PC-FW	PC-SW
Tau	0.17 ± 0.00 a	0.14 ± 0.00 a
Asp ¥	8.80 ± 0.06 a	8.51 ± 0.05 a
Glu ¥	13.46 ± 0.08 a	13.21 ± 0.08 a
Gly ¥	3.70 ± 0.02 a	3.98 ± 0.02 a
Ala ¥	4.89 ± 0.04 a	4.84 ± 0.03 a
Thr *	3.38 ± 0.03 a	3.26 ± 0.01 a
Val *	3.91 ± 0.02 a	3.77 ± 0.01 a
Met *	1.69 ± 0.01 a	1.43 ± 0.00 a
Ile *	3.81 ± 0.04 a	3.69 ± 0.01 a
Leu *	6.70 ± 0.03 a	6.44 ± 0.04 a
Phe *	3.59 ± 0.02 a	3.48 ± 0.03 a
Lys *	5.88 ± 0.02 a	6.23 ± 0.07 b
Trp *	1.44 ± 0.05 a	1.94 ± 0.07 b
His	1.94 ± 0.01 a	1.78 ± 0.01 a
Arg	7.99 ± 0.05 a	7.48 ± 0.04 a
Ser	3.42 ± 0.01 a	3.18 ± 0.03 a
Tyr	3.14 ± 0.03 a	3.01 ± 0.02 a
Pro	2.58 ± 0.01 a	2.42 ± 0.01 a
Cys	0.53 ± 0.09 a	0.66 ± 0.05 a
∑TAA	80.85 ± 0.33 a	79.29 ± 0.48 a
∑TDAA	30.85 ± 0.21 a	30.54 ± 0.19 a
∑TDAA/∑TAA (%)	38.16 ± 0.10 a	38.52 ± 0.00 a
∑EAA	30.41 ± 0.13 a	30.23 ± 0.25 a
∑NEAA	40.51 ± 0.14 a	39.81 ± 0.18 a
∑EAA/∑TAA (%)	37.61 ± 0.01 a	38.12 ± 0.08 a
∑EAA/∑NEAA (%)	75.07 ± 0.07 a	79.93 ± 0.28 a
∑BCAA	14.42 ± 0.09 a	13.90 ± 0.06 a
∑AAC	6.73 ± 0.05 a	6.49 ± 0.05 a
∑BCAA/∑AAC	2.14 ± 0.00 a	2.14 ± 0.01 a

Note: TAA means total amino acids, TDDA means total delicious amino acids, EAA means essential amino acids, NEAA means no essential amino acids, BCAA means branched chain amino acids, and AAC means aromatic amino acids. Amino acids with ¥ and * represent delicious amino acids and essential amino acids, respectively. Different letters on the same column of data indicate significant differences (*p* < 0.05).

**Table 4 foods-14-01997-t004:** Composition and content of essential amino acids.

Amino Acids	FAO/WHO Standard Mode	Whole Egg Mode	PC-FW			PC-SW		
EEA	Content (mg/g)	Content (mg/g)	Content (mg/g)	AAS	CS	Content (mg/g)	AAS	CS
Thr	250	292	211	0.85	0.72	204	0.82	0.70
Val	310	411	244	0.79	0.55	236	0.76	0.53
Met+Cys	220	386	139	0.63	0.36	136	0.59	0.34
Ile	250	331	238	0.95	0.72	231	0.92	0.70
Leu	440	534	419	0.95	0.78	403	0.91	0.75
Phe+Tyr	380	565	421	1.11	0.74	406	1.07	0.72
Lys	340	441	368	1.08	0.83	389	1.15	0.88
Trp	60	99	90	1.50	0.91	121	2.02	1.22
Total	2250	3059	2130	7.86	5.61	2126	8.24	5.84
EAAI			68.02			69.01		

**Table 5 foods-14-01997-t005:** Analysis of nucleotides and betaine in crayfish (fresh weight, μg/g).

Indexes	PC-FW	PC-SW
Betaine	7691.30 ± 51.60 ^a^	8161.60 ± 138.20 ^b^
Geosmin	(4.32 ± 0.09) × 10^−3 a^	(3.13 ± 0.09) × 10^−3 b^
CMP	71.70 ± 0.41 ^a^	172.20 ± 2.87 ^b^
UMP	14.28 ± 0.13 ^a^	19.30 ± 0.42 ^b^
IMP	1283.69 ± 43.79 ^a^	2835.06 ± 63.27 ^b^
GMP	ND	20.52 ± 0.07

Note: ND means not detected, and different superscripts on the same column of data indicate significant differences (*p* < 0.05).

## Data Availability

The original contributions presented in the study are included in the article/Appendix A, further inquiries can be directed to the corresponding authors.

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
