# Peer review of "A Comparative Analysis on the Biochemical Composition and Nutrition Evaluation of Crayfish (Procambarus clarkii) Cultivated in Saline-Alkali and Fresh Water"

_foods, 2025, doi:10.3390/foods14111997_

Round 1
Reviewer 1 Report
Comments and Suggestions for Authors
Thank you for the opportunity to participate in the revision of the article entitled “A comparative analysis on the biochemical composition and nutrition evaluation of crayfish (Procambarus clarkii) cultivated in saline-alkali and fresh water”.
The objective of this study is to investigate the nutritional content and qualitative properties of crayfish cultivated from saline-alkali and fresh water systematically.
I think the topic is interesting and is covered discreetly. The tables and graphs help the reader to understand the results obtained. The bibliographical references are also adequate for the topic. However, the section that requires more editing is the 'Materials and Methods' section. In many cases, the authors indicated that some methods referred to studies already in the literature. While this may be acceptable, it would be preferable to provide a brief description of the methods employed, including whether they have been analytically validated. Additionally, the specifics of the instruments used (model, brand and instrumental parameters) are not provided. I therefore suggest that this section be extensively revised.
My assessment is 'major revisions'. The following comments have been provided:
1. Introduction
Lines 36-37: “Procambarus clarkii, commonly known as freshwater crayfish, belongs to the Crustacea order, Decapoda, Cambaridae family, and Procambarus genus”. Scientific names should be in italics. I therefore recommend that you edit the text.
2. Materials and Methods
2.3. Determination of nutritional components in crayfish
Lines 105-108: “Nutritional components are reflected by water content, crude protein content, crude fat content, and ash in this study based on the method described in previous study [18]. All the tested indexes of nutritional components of crayfish were followed by Khan et al. [18]”. I understand that you have referred to the methods given in Bibliography 18. However, I suggest that you still include a brief explanation of the methods used.
2.4. Determination of trace elements in crayfish
The above applies. Describe the method used. Which instrument did you use? How did you pretreat the samples? Has the method been validated? Why did you not consider determining the content of toxic and potentially toxic elements?
2.6. Analysis of amino acid composition in crayfish
Lines 135-137: “The mixture was filtered through a 0.45 μm membrane and tested for amino acids using an amino acid analyzer”. I would recommend specifying the instrumentation used.
2.7. Analysis of fatty acid composition in crayfish
Line 138: Change “crayfis” to “crayfish”.
Lines 141-142: “Following transesterification at 70 °C for 1 h, approximately 2 mL of the supernatant was mixed with 0.75 mL of ddH2O and 2 mL”. What is the quantity of 2 mL in question?
Lines 142-144: “A total of 1 mL of filtered sample through a 0.45 μm membrane was used for fatty acid composition determination via gas chromatography”. I would recommend specifying the instrumentation used.
2.8. Determination of betaine in crayfish
Lines 153-154: “The mobile phase was acetonitrile: water (60:40) and the detection wavelength was 195 nm”. From what you have written, I believe that you used an HPLC. However, I suggest indicating this in the text and reporting the model and company.
2.9. Determination of nucleotides in crayfish
Lines 159-162: “The supernatants were blended, neutralized (pH 6.5) with 1 mol/L KOH, fixed to 50 mL with ddH2O, percolated through a 0.45 μm filter membrane, and eventually analyzed by high-performance liquid chromatography”. I suggest indicating the model and company of the instrument.
3. Results and discussion
3.1. Nutritional profiles of crayfish
Lines 218-219: “Similar findings have been reported in studies on Lateolabrax japonicus and Micropterus salmoides, as well as in Scophthalmus maximus, where the crude fat content decreased as salinity increased [30-32]”. The following species should be reported in italics: Lateolabrax japonicus, Micropterus salmoides and Scophthalmus maximus.
3.3. Trace elements in crayfish
Lines 240-242: “The results of contents of trace elements in PC-SW and PC-FW showed that the content of Cu in PC-FW (34.68 ± 0.30 mg/kg) was higher than that in PC-SW (shown in Figure 2)”. Report the copper value for PC-SW, too.
Lines 243-246: “Significant differences were found on the concentrations of Fe, Zn, Se, Na, Ca, Mg, and Mn in PC-SW, with respective concentration of 102.49 ± 1.69 mg/kg, 95.45 ± 2.00 mg/kg, 2.49 ± 0.04 mg/kg, 18.29 ± 0.06 g/kg, 8.66 ± 0.01 g/kg, 6.09 ± 0.01 g/kg, 2.28 ± 0.01 mg/kg, and 18.38 ± 0.20 mg/kg”. Are the significant differences with respect to PC-FW? In that case, I recommend reporting the concentration for this sample as well.
3.4. Fatty acids profiles in crayfish
Lines 265-268: “The results showed that the total SFA content in PC-SW (32.72%) was significantly greater than that (29.68%) in PC-FW (p <0.05), while the total PUFA content in PC-FW showed higher values at 39.12%, which was reflected in the enrichment of n-3 fatty acids (such as C18:3n3) and arachidonic acid (ARA)”. I recommend standardizing how fatty acids are referred to. Either use the acronym or the name.
3.7. Betaine, geosmin and nucleotides content in crayfish
Line 374: “Geosmin is mainly synthesized by actinomycetes such as Streptomyces sp. [44]”. Report: 'Streptomyces sp.' in italics.
Author Response
Dear reviewer,
Thanks so much for your comments, we have modified the manuscript point to point based on your comments. All the modified parts are marked in yellow. Hopefully, our revised manuscript could satisfy your questions.
1. Introduction
Lines 36-37: “Procambarus clarkii, commonly known as freshwater crayfish, belongs to the Crustacea order, Decapoda, Cambaridae family, and Procambarus genus”. Scientific names should be in italics. I therefore recommend that you edit the text.
A: Thanks for your comments, we have changed the scientific names and put them in italics.
2.Materials and Methods
2.3. Determination of nutritional components in crayfish
Lines 105-108: “Nutritional components are reflected by water content, crude protein content, crude fat content, and ash in this study based on the method described in previous study [18]. All the tested indexes of nutritional components of crayfish were followed by Khan et al. [18]”. I understand that you have referred to the methods given in Bibliography 18. However, I suggest that you still include a brief explanation of the methods used.
A:Thanks for your comments, we have added more information including a brief explanation of the methods we used in lines 124-130.
3. 2.4. Determination of trace elements in crayfish
The above applies. Describe the method used. Which instrument did you use? How did you pretreat the samples? Has the method been validated? Why did you not consider determining the content of toxic and potentially toxic elements?
A:Thanks for your comments. we have added the description how we treated the crayfish samples in lines 107-110. and We added how we conduct the detection of trace elements in crayfish in lines 135-145. In this study, we focused on the trace elements like K, Ca, Fe, Se and so on because they participate enzymatic activity for human body, this is the main goal for the study. therefore, we did not consider determining the toxic or the potential elements.
4. 2.6. Analysis of amino acid composition in crayfish
Lines 135-137: “The mixture was filtered through a 0.45 μm membrane and tested for amino acids using an amino acid analyzer”. I would recommend specifying the instrumentation used.
A: Thanks for your comments. we have specified the instrumentation used in our study in line 168 as LA8080, Hitachi Co., Tokyo, Japan.
5. 2.7. Analysis of fatty acid composition in crayfish
Line 138: Change “crayfis” to “crayfish”.
A: sorry for that mistake, we have changed it as crayfish in line 169.
6.Lines 141-142: “Following transesterification at 70 °C for 1 h, approximately 2 mL of the supernatant was mixed with 0.75 mL of ddH2O and 2 mL”. What is the quantity of 2 mL in question?
A: Sorry that some key information is missing, we have added the information in lines 173-179 related to this question.
7.Lines 142-144: “A total of 1 mL of filtered sample through a 0.45 μm membrane was used for fatty acid composition determination via gas chromatography”. I would recommend specifying the instrumentation used.
A: Thanks for your comments, we have specified the instrument used for amino acid analysis in lines 175-176..
8. 2.8. Determination of betaine in crayfish
Lines 153-154: “The mobile phase was acetonitrile: water (60:40) and the detection wavelength was 195 nm”. From what you have written, I believe that you used an HPLC. However, I suggest indicating this in the text and reporting the model and company.
A: Thanks for your comments, we used the HPLC to determine the betaine in our study, therefore, we added this information in lines 188-189, with the indication of the HPLC company we used.
9. 2.9. Determination of nucleotides in crayfish
Lines 159-162: “The supernatants were blended, neutralized (pH 6.5) with 1 mol/L KOH, fixed to 50 mL with ddH2O, percolated through a 0.45 μm filter membrane, and eventually analyzed by high-performance liquid chromatography”. I suggest indicating the model and company of the instrument.
A: we appreciate your concerns, again we double checked the gear we used in our study, and indicated the model and company of the instrument in our study. Those information were added in lines 199-200.
10.Results and discussion
3.1. Nutritional profiles of crayfish
Lines 218-219: “Similar findings have been reported in studies on Lateolabrax japonicus and Micropterus salmoides, as well as in Scophthalmus maximus, where the crude fat content decreased as salinity increased [30-32]”. The following species should be reported in italics: Lateolabrax japonicus, Micropterus salmoides and Scophthalmus maximus.
A: Thanks for your suggestion, we have changed the species in italics. They are now marked in manuscript at lines 253-254.
11. Trace elements in crayfish
Lines 240-242: “The results of contents of trace elements in PC-SW and PC-FW showed that the content of Cu in PC-FW (34.68 ± 0.30 mg/kg) was higher than that in PC-SW (shown in Figure 2)”. Report the copper value for PC-SW, too.
A:Thanks for your comments, we have added the value of Cu in PC-SW as 32.48±0.22 mg/kg in line 277.
12. Lines 243-246: “Significant differences were found on the concentrations of Fe, Zn, Se, Na, Ca, Mg, and Mn in PC-SW, with respective concentration of 102.49 ± 1.69 mg/kg, 95.45 ± 2.00 mg/kg, 2.49 ± 0.04 mg/kg, 18.29 ± 0.06 g/kg, 8.66 ± 0.01 g/kg, 6.09 ± 0.01 g/kg, 2.28 ± 0.01 mg/kg, and 18.38 ± 0.20 mg/kg”. Are the significant differences with respect to PC-FW? In that case, I recommend reporting the concentration for this sample as well.
A: Thanks for your helpful comments, we have reported the concentration of different trace elements in PC-FW as they are shown in lines 282-285.
13.3.4. Fatty acids profiles in crayfish
Lines 265-268: “The results showed that the total SFA content in PC-SW (32.72%) was significantly greater than that (29.68%) in PC-FW (p <0.05), while the total PUFA content in PC-FW showed higher values at 39.12%, which was reflected in the enrichment of n-3 fatty acids (such as C18:3n3) and arachidonic acid (ARA)”. I recommend standardizing how fatty acids are referred to. Either use the acronym or the name.
A: Thanks for your helpful comments, we have standardized the name of fatty acids in our manuscript. The revised part are marked in yellow.
14. 3.7. Betaine, geosmin and nucleotides content in crayfish
Line 374: “Geosmin is mainly synthesized by actinomycetes such as Streptomyces sp. [44]”. Report: 'Streptomyces sp.' in italics.
A: Thanks for your comments, the Streptomyces are written in italics.
Reviewer 2 Report
Comments and Suggestions for Authors
The manuscript “A comparative analysis on the biochemical composition and nutrition evaluation of crayfish (Procambarus clarkii) cultivated in saline-alkali and fresh water” addresses an interesting topic, the comparison between freshwater and saline-alkali cultivation is timely and valuable for aquaculture development in marginal lands. However, the article shows serious weaknesses in the discussion of its results.
General comments
Many result subsections (e.g., 3.2, 3.3, 3.5, 3.7) merely restate data without sufficient interpretation or connection to the broader context.
Discussion should address: Why does salinity enhance certain parameters? How do the findings compare with other species?
Scientific names like Procambarus clarkii must be italicized.
Some tables and equations have formatting issues (e.g., "rpm" should be converted to relative centrifugal force (×g)).
Table 4 is mislabeled as Table 1 in the text.
Attached manuscript with observations

Author Response
Dear reviewer,
Thanks so much for your comments, we have modified the manuscript point to point based on your comments. All the modified parts are marked in yellow. Hopefully, our revised manuscript could satisfy your questions.
1.Many result subsections (e.g., 3.2, 3.3, 3.5, 3.7) merely restate data without sufficient interpretation or connection to the broader context. Discussion should address: Why does salinity enhance certain parameters? How do the findings compare with other species?
A:Thanks for your helpful comments, we have added relative data with some discussion to connect between results, hopefully our modification could satisfy your concerns.
2. Scientific names like Procambarus clarkii must be italicized.
A:Thanks for your comments, we have made the modification that keeps the scientific names in italics.
3.Some tables and equations have formatting issues (e.g., "rpm" should be converted to relative centrifugal force (×g))
A:Thanks for your comments, we have double checked the mistakes, and changed them. the modified parts are marked in yellow.
4. Table 4 is mislabeled as Table 1 in the text.
A: Thanks for your comments, we have changed the mislabeled table.
5. Attached manuscript with observations
A:Thanks for your comments, we have put the full name for BCAA, NEAA, EAA for the first time. For the discussion part at very beginning, we described the importance of this study, and why should conduct the study, therefore, i think it is important to keep the first two paragraphs in the discussion parts. at last, thanks again, we have added more discussion based on your comments.
Reviewer 3 Report
Comments and Suggestions for Authors
This is an original and innovative study that presents a comparative analysis of the biochemical composition and nutritional value of Procambarus clarkii cultivated in two contrasting environments: freshwater and saline-alkali water. The study evaluates relevant parameters such as amino acid profiles, fatty acids, and mineral content to assess the effect of culture conditions on the organism's nutritional quality. The manuscript is well-structured and thoughtfully presented; however, several revisions are necessary prior to its consideration for publication in Foods, an MDPI journal. Below, I offer a series of comments intended to enhance the content and strengthen its scientific contribution.
1.- The abstract provides a general overview of the study’s objective and highlights the main findings regarding the nutritional composition of Procambarus clarkii cultivated in freshwater and saline-alkali environments. However, the writing could be improved for greater clarity and readability. Specifically, the sentence spanning lines 13 to 18 is overly long and dense, and it is recommended to divide it into at least two shorter, more concise sentences. Additionally, the abstract would benefit from a final statement that emphasizes the practical relevance of the findings, which would help strengthen the impact and contribution of the study. Lastly, all abbreviations used in the abstract—such as NEAA, EAA, BCAA, AAC, and EAAI —should be defined upon first mention to ensure clarity for a broader scientific audience.
2.- he introduction provides a general context for the study, highlighting the economic importance of Procambarus clarkii and the potential of saline-alkali environments for its cultivation. However, it presents some important weaknesses that should be addressed to strengthen the scientific rationale. First, the logical flow is limited; the text transitions too abruptly from the species’ commercial relevance to the cultivation environment without sufficiently explaining why saline-alkali conditions are emerging as a research or production focus. Additionally, in lines 75–77, the authors state that several studies have analyzed the nutritional composition of crayfish muscle, yet no references are provided to support this claim. It is strongly recommended to include at least two recent and relevant citations to substantiate this statement. Finally, the objective of the study should be clearly articulated at the end of the introduction, with a concise sentence that outlines the comparative approach and its scientific or practical relevance.
Line 36 Procambarus clarkii in italics
3.- The Materials and Methods section provides a general overview of the experimental procedures, including sample collection, biochemical analyses, and statistical processing. However, several improvements are necessary to enhance transparency and reproducibility. In lines 86–88, the authors mention the collection of crayfish samples but do not indicate the size and weight of the organisms, which are critical parameters for nutritional comparison; this information should be clearly stated. Furthermore, the description of the environmental conditions of the saline-alkali and freshwater ponds is vague; it is essential to include detailed parameters such as salinity, pH, temperature, and duration of cultivation. In line 149, the authors report centrifugation conditions using rpm, but this unit should be replaced with relative centrifugal force (×g) to ensure clarity and reproducibility across different equipment. In addition, it is important to include detailed specifications of all instruments used (e.g., model, manufacturer, city, and country) to enhance methodological transparency. In line 184, the authors mention the use of SPSS software for statistical analysis; they should indicate the software version as well as the company name, city, and country. Lastly, the manuscript should clearly state whether ethical approval was obtained for the use of live organisms in the study; the authors are kindly requested to provide the corresponding ethics committee approval certificate.
4.- The discussion section lacks sufficient analytical depth, as it is primarily limited to a descriptive comparison of the quantitative differences observed between the two culture environments, without exploring the physiological or environmental mechanisms that may underlie these variations. Moreover, there is a noticeable absence of comparison between the results of the present study and findings reported in the scientific literature, which hinders the contextualization of the data within the current state of knowledge. It is strongly recommended to incorporate up-to-date and relevant references that support or contrast the findings. Additionally, the technological and economic relevance of the results should be emphasized, as doing so would significantly enhance the applied value of the study.
5.- The conclusions are overly general and do not accurately or quantitatively reflect the results obtained. Furthermore, the practical applicability of the findings is not discussed. It is recommended that the conclusions be restructured to align with the main findings and the objective of the present study, highlighting the most relevant results based on the data presented and linking them to potential applications in aquaculture, nutrition, or product development. Additionally, it would be advisable to include a critical reflection on the study’s limitations and specific suggestions for future research.
Comments on the Quality of English Language The manuscript is generally understandable; however, a thorough language revision by a native English speaker is recommended to improve clarity, grammar, and flow throughout the text.Author Response
Dear reviewer,
Thanks so much for your comments, we have modified the manuscript point to point based on your comments. All the modified parts are marked in yellow. Hopefully, our revised manuscript could satisfy your questions.
1.- The abstract provides a general overview of the study’s objective and highlights the main findings regarding the nutritional composition of Procambarus clarkii cultivated in freshwater and saline-alkali environments. However, the writing could be improved for greater clarity and readability. Specifically, the sentence spanning lines 13 to 18 is overly long and dense, and it is recommended to divide it into at least two shorter, more concise sentences. Additionally, the abstract would benefit from a final statement that emphasizes the practical relevance of the findings, which would help strengthen the impact and contribution of the study. Lastly, all abbreviations used in the abstract—such as NEAA, EAA, BCAA, AAC, and EAAI —should be defined upon first mention to ensure clarity for a broader scientific audience.
A: Thanks so much for your helpful comments, we have shorten the sentence in lines 13-18. In addition, we highlighted some important findings in lines 30-31. At last, we gave the full name for all the abbreviations for the first use.
2.- he introduction provides a general context for the study, highlighting the economic importance of Procambarus clarkii and the potential of saline-alkali environments for its cultivation. However, it presents some important weaknesses that should be addressed to strengthen the scientific rationale. First, the logical flow is limited; the text transitions too abruptly from the species’ commercial relevance to the cultivation environment without sufficiently explaining why saline-alkali conditions are emerging as a research or production focus. Additionally, in lines 75–77, the authors state that several studies have analyzed the nutritional composition of crayfish muscle, yet no references are provided to support this claim. It is strongly recommended to include at least two recent and relevant citations to substantiate this statement. Finally, the objective of the study should be clearly articulated at the end of the introduction, with a concise sentence that outlines the comparative approach and its scientific or practical relevance.
Line 36 Procambarus clarkii in italic
A: Thanks for your comments, based on your suggestion, we have added updated results in the introduction part to support our idea, and we also added new statement to support the objective of our study. at last, we rewrote the Procambarus clarkii in italic.
3.- The Materials and Methods section provides a general overview of the experimental procedures, including sample collection, biochemical analyses, and statistical processing. However, several improvements are necessary to enhance transparency and reproducibility. In lines 86–88, the authors mention the collection of crayfish samples but do not indicate the size and weight of the organisms, which are critical parameters for nutritional comparison; this information should be clearly stated. Furthermore, the description of the environmental conditions of the saline-alkali and freshwater ponds is vague; it is essential to include detailed parameters such as salinity, pH, temperature, and duration of cultivation. In line 149, the authors report centrifugation conditions using rpm, but this unit should be replaced with relative centrifugal force (×g) to ensure clarity and reproducibility across different equipment. In addition, it is important to include detailed specifications of all instruments used (e.g., model, manufacturer, city, and country) to enhance methodological transparency. In line 184, the authors mention the use of SPSS software for statistical analysis; they should indicate the software version as well as the company name, city, and country. Lastly, the manuscript should clearly state whether ethical approval was obtained for the use of live organisms in the study; the authors are kindly requested to provide the corresponding ethics committee approval certificate.
A: Thank you so much for your comments. first of all, we put the ethics statement in MM section as it shown in 2.1 in lines 90-92. secondly, we also added the weight and length of each juvenile crayfish in line 94, " 2.0±0.1 cm in length and 5.0 ±0.3 in weight." we also mentioned how we changed the water, and how we tested the pH, salinity, and alkalinity (which is also shown in supplementary materials). Thirdly, we changed all relative centrifugal force from rpm to g to ensure the clarity. In addition, we also modified the methodological procedures in detail such as in fatty acid composition in lines 173-179, and nutritional profiles and trace elements in lines 124-130 and 135-145, respectively. As we added those information in the manuscript, we also specified the the name, brand, city and country of the instruments used in the manuscript as we as the SPSS software.
4.- The discussion section lacks sufficient analytical depth, as it is primarily limited to a descriptive comparison of the quantitative differences observed between the two culture environments, without exploring the physiological or environmental mechanisms that may underlie these variations. Moreover, there is a noticeable absence of comparison between the results of the present study and findings reported in the scientific literature, which hinders the contextualization of the data within the current state of knowledge. It is strongly recommended to incorporate up-to-date and relevant references that support or contrast the findings. Additionally, the technological and economic relevance of the results should be emphasized, as doing so would significantly enhance the applied value of the study.
A:Thanks for your helpful comments, we tried our best to modified the results and discussion part in corporate up-to-date results from relevant publications. The revised version was marked in yellow. hopefully our reply could satisfy your concerns.
5.- The conclusions are overly general and do not accurately or quantitatively reflect the results obtained. Furthermore, the practical applicability of the findings is not discussed. It is recommended that the conclusions be restructured to align with the main findings and the objective of the present study, highlighting the most relevant results based on the data presented and linking them to potential applications in aquaculture, nutrition, or product development. Additionally, it would be advisable to include a critical reflection on the study’s limitations and specific suggestions for future research.
A:Thanks for your comments, we added new conclusion related to your concerns in lines 471-475.
Reviewer 4 Report
Comments and Suggestions for Authors
Dear author,
Thank you for your work
My general comments
Please check scientific name format in whole manuscript
Check H2O format for whole manuscript
L138 check the name of crayfish for whole manuscript
My detail comments
Title: accepted as it reflects the whole content of the manuscript
Abstract:
Overall is accepted. but it would better to provide value of the all mentioned results
Keywords
it would be better to avoid use same words as in the title
Introduction
well written. but it would be better to provide more literature review on application of saline alkaline water in aqua farming together with their outcomes in line with the current study
Material and methods
it would be better to provide an ethical statement in the beginning of this section
L86: weight and lenght of the juvenile crayfish? suggest to provide on how to measure water quality, frequency? and explain water source for this experiment
2.3 & 2.4: please explain more detail
please explain on how the crayfish were killed?
L182: please clarify if any data normality and HOV tests?
Result and discussion
authors able to perform a good result presentation
However, the discussion is not comprehensive. Not all findings were discusses
therefore, suggest to make sure each finding is discuss in detail with the current updated references
conclusion
Please provide research gap, future work and the ideal dose of saline alkaline water for crayfish farming
Author Response
Dear reviewer,
Thanks so much for your comments, we have modified the manuscript point to point based on your comments. All the modified parts are marked in yellow. Hopefully, our revised manuscript could satisfy your questions.
1.Please check scientific name format in whole manuscript, Check H2O format for whole manuscript. L138 check the name of crayfish for whole manuscript.
A: Thanks for your comments, we have went through the whole manuscript, and tried to make sure the scientific name, H20, and the crayfish are correct in the manuscript.
2. Title: accepted as it reflects the whole content of the manuscript.
A: Thank you so much.
3.Abstract: Overall is accepted. but it would better to provide value of the all mentioned results.
A: Thank you for your comments, we have provide the results in the abstract section in lines 29-30.
4.Keywords: it would be better to avoid use same words as in the title.
A: Thanks for your comments, we have added 2 more keywords as "Amino acid composition; Flavor compounds" in line 34.
5.Introduction: well written. but it would be better to provide more literature review on application of saline alkaline water in aqua farming together with their outcomes in line with the current study.
A: Thank you for your comments. We have provide some information on the application of saline alkaline water aquaculture farming and outcomes on weight, survival rates in lines 59-61.
6. Material and methods: it would be better to provide an ethical statement in the beginning of this section
A: Thanks for you comments, in fact we have the ethical statement and we have added this information in the MM section. as "Animal experiments were approved by the Ethics Committee of the East China Sea Fisheries Research Institute, Chinese Academy of Fishery Sciences (Approval code: 2024-12-ZX-013). in lines 90-92.
7. L86: weight and lenght of the juvenile crayfish? suggest to provide on how to measure water quality, frequency? and explain water source for this experiment.
A: Thanks for your comments, we have provided the weight and length of the juvenile crayfish in line 94 as "Juvenile crayfish (2.0 ± 0.1 cm in length, 5.0 ± 0.3 g in weight) " The water changes and water source are described in lines 99-103 as "Each aquaculture farm underwent a water change from the local well water every 15 days, with each change typically accounting for a quarter of the total farm water volume, to avoid excessive fluctuations in water quality that may affect crayfish growth. After each water change, the salinity, alkalinity, and pH were measured accordingly. "
8. 2.3 & 2.4: please explain more detail. please explain on how the crayfish were killed?
A: Thanks for your comments, we rewrote the MM section in order to explain what we have done in detail. we also added more information on how we killed the crayfish and what we have done in the study in lines 107-109.
9. L182: please clarify if any data normality and HOV tests?
A:Thanks for your comments, we randomly picked up the lots samples to ensure the adequacy of data, at the same time, we used ANOVA analysis, which requires data to meet normality and homogeneity of variance, to ensure the reliability of data. That's why we didnot clarify the procedures.
10. Result and discussion
authors able to perform a good result presentation
However, the discussion is not comprehensive. Not all findings were discusses
therefore, suggest to make sure each finding is discuss in detail with the current updated references
A: Thanks for your comments, we have added information in the discussion part to make it comprehensive.
11. conclusion
Please provide research gap, future work and the ideal dose of saline alkaline water for crayfish farming .
A:Thanks for your comments, it is really useful that to offer the future work on the ideal dose of saline-alkali water for crayfish farming.
Round 2
Reviewer 1 Report
Comments and Suggestions for Authors
The authors made almost all of the requested changes. However, I have not received a response regarding my previous comment on the validation of the ICP-OES method.
Therefore, I suggest that the authors indicate whether the method has been validated in terms of linearity, sensitivity and accuracy, and provide a table showing this.
Author Response
Dear reviewer,
Thanks for your comments, we have modified the manuscript based on your suggestions point to point carefully.
1.The authors made almost all of the requested changes. However, I have not received a response regarding my previous comment on the validation of the ICP-OES method.
Therefore, I suggest that the authors indicate whether the method has been validated in terms of linearity, sensitivity and accuracy, and provide a table showing this.
A: thanks for your comments, we have made a table for method validation results including standard curve, LOD, RSD, and put the table in Table S2.
Reviewer 2 Report
Comments and Suggestions for Authors
The manuscript has improved considerably, but there are still minor details that need to be corrected.
I have left comments in the attached document.

Author Response
Dear Reviewer,
Thanks so much for your comments, we have modified the manuscript based on your comments point to point carefully,
1.Line 129. Changed to using 5.00 ± 0.05 g of crayfish.
A: Thanks for your comments, we have changed it.
2.Line 131-132. change to approximately 1.20 ± 0.03 g of crayfish.
A: thanks for your comments, we have changed it.
3.Line 141-142. changed the 0.50 ± 0.01 g of crayfish
A: thanks for your comments, we have changed it.
4. Line 171 changed to about 30.00 ± 0.01 mg of sample
A: Thanks for your comments, we have changed it.
5. This paragraphy does not describe the results of the study.
A: Thanks for your comments, we have deleted those two paragraphes.
6. Line267 Lazarevic et al. found .... add the ref. number
A: Thanks for your suggestion, we have added the reference number [28] at the end of the sentence.
7.Line 269. Sun et al . found that.... add the ref. number
A: Thanks for your comments, we have added the reference number [29] at the end of the sentence.
8. please review and cite appropriately in the rest of the manuscript.
A: Thanks for your comments, we have went through the manuscript and made sure all the citations are in correct numbers.
Reviewer 3 Report
Comments and Suggestions for Authors
Upon evaluating the revised version, I commend the authors for their substantial effort in addressing all of my previous comments. Their thoughtful revisions are greatly appreciated.
However, I recommend that the manuscript undergo a final language review by a native English speaker or professional editor, as several grammatical and stylistic issues remain that could affect the clarity and readability of the work.
Comments on the Quality of English Language
I recommend that the manuscript undergo a final language review by a native English speaker or professional editor.
Reviewer 4 Report
Comments and Suggestions for Authors
Dear author,
Thank you for your revision
My general comments:
will recommend after minor revision
My detail comments:
There are several issues:
Abstract:
it would be better to provide the result is significant or not? together with its actual p value
Introduction
Please provide market value of the crayfish
Materials and methods
explain in detail on how to measure water quality and water source for the study
results
Please check all tables and figures format. make sure abbreviation use is in correct form
Author Response
Dear Reviewer,
Thank you again for your helpful comments, we have modified the manuscript point to point carefully. Hopefully our revised manuscript could satisfy your concerns.
1.Abstract: it would be better to provide the result is significant or not? together with its actual p value
A: Thanks for your comments, we have provided the p values for the results that present significantly in lines 19, 22 and 30.
2.Introduction: Please provide market value of the crayfish
A: Thanks for your comments, it is very useful information, we have added a sentence as "In 2003, the Chinese market for crayfish was valued at roughly 68.2 billion US dollars [3,4]." in lines 44-45.
3. Materials and methods: explain in detail on how to measure water quality and water source for the study.
A: Thanks for your comments, we have added the information in lines 103-110.
4.results: Please check all tables and figures format. make sure abbreviation use is in correct form.
A: Thanks for your comments, we have double checked all the tables and figures to make sure every abbreviation are correct.